# Prioritizing Patient Safety: Analysis of the Procurement Process of Infusion Pumps in Spain

**DOI:** 10.3390/ijerph20247179

**Published:** 2023-12-14

**Authors:** Laura Herrero, Blanca Sánchez-Santiago, Marina Cano, Ramon Sancibrian, Raj Ratwani, Galo Peralta

**Affiliations:** 1Innovation Support Unit, Instituto de Investigación Marqués de Valdecilla-IDIVAL, 39011 Santander, Spain; innovacion2@idival.org (L.H.); innovacion5@idival.org (M.C.); 2Clinical Pharmacology Service, Hospital Universitario Marqués de Valdecilla-IDIVAL, 39008 Santander, Spain; mblanca.sanchezs@scsalud.es; 3Department of Structural and Mechanical Engineering, Universidad de Cantabria, 39005 Santander, Spain; sancibrr@unican.es; 4MedStar Health Research Institute, Washington, DC 20010, USA; raj.m.ratwani@medstar.net

**Keywords:** procurement, patient safety, usability, infusion pumps

## Abstract

To understand whether patient safety and human factors are considered in healthcare technology procurement, we analyzed the case of infusion pumps as their use critically affects patient safety. We reviewed infusion pump procurements in the Spanish Public Sector Procurement Database. Sixty-three batches in 29 tenders for supplying 12.224 volumetric and syringe infusion pumps and consumables for an overall budget of EUR 30.4 M were identified and reviewed. Concepts related to “ease of use” were identified in the selection requirements of 35 (55.6%) batches, as part of the criteria for the selection of pumps in 23 (36.5%) batches, related to “intuitiveness” in the selection requirements of 35 (55.6%) batches, and in the criteria in 10 (15.9%) batches. No method to evaluate the ease of use, intuitiveness, or usability was mentioned. A review of the procurement teams responsible for the evaluation of the tenders showed no reported human factors or patient safety expertise. We conclude that infusion pump procurement considers usability as a relevant criterion for selection. However, no human factor experts nor specific methods for evaluation of the technology in this field are usually defined. Potential room for refining the selection of healthcare technology to improve patient safety is detected.

## 1. Introduction

Patient safety is a global priority and a central focus of the World Health Organization [1], as well as other oversight agencies [2,3]. There are many facets to patient safety, and one critical aspect is the usability of medical devices [4], health information technology [5], and other products, which is the extent to which the technology or product can be used effectively, efficiently, and satisfactorily [6]. Products that are poorly designed, developed, and implemented can have poor usability, which can directly impact patient safety by resulting in errors that harm patients [7]. For example, an inaccurately programmed infusion pump due to a confusing display can result in a patient receiving the wrong amount of medication, resulting in an over- or underdose [8,9,10,11].

Human factors, a multidisciplinary science focused on understanding human capabilities and designing tools and technologies to meet these capabilities, are instrumental in promoting usable and safe medical devices and technologies [7]. Human factor methods such as direct observation, interviews, and surveys to gather user needs, rigorous user evaluation, usability testing, and heuristic evaluations all serve to improve product usability [12,13]. In certain countries, there are oversight agencies, such as the United States Food and Drug Administration (FDA), that require usability testing for medical devices and other products before introduction to the market [14].

In Europe, a key step to assessing the usability and safety of medical devices is the public procurement process [15]. Currently, public procurement in Europe accounts for approximately 14% of gross domestic product and is an essential vehicle for implementing government policies and meeting national strategic objectives, as well-functioning public procurement markets contribute to improving the competitiveness of quality service strategies. Identifying usability and patient safety issues during procurement can prevent patient harm and can serve to improve medical products [16].

In this article, we review public procurements in Spain to identify whether usability and human factors were taken into consideration during the procurement process. We focused specifically on infusion pumps, given the prevalence of these devices and the importance of usability for the safe use of pumps. Infusion pumps are recognized as devices frequently involved in medication errors. From 2005 to 2009, the FDA received approximately 56,000 adverse event reports associated with the use of infusion pumps, including injuries and deaths. Manufacturers made 87 infusion pump recalls addressing identified safety concerns during this period. Seventy of these recalls were designated as Class II, which implies they are likely to cause temporary or medically reversible adverse health consequences. Fourteen recalls were Class I, which means they are likely to cause serious adverse health consequences or death [17]. Other studies have identified a high rate of error in the administration of intravenous medications with smart pumps, with relatively few potentially harmful errors [18,19,20]. Infusion pumps provide an elevated level of control, accuracy, and precision in medication administration, and reduce certain types of medication errors, resulting in improved patient care [17,18,19,20,21]. At the same time, infusion pumps have been associated with persistent safety issues that can lead to over- or under-infusion and overlooked or delayed therapy [22]. Despite the growing support for the use of smart pumps as an element of safety strategies, the literature shows that user error, incorrect programming, and equipment failures continue to occur [23]. Several strategies have been proposed for mitigating infusion pump safety problems as addressing known problems training and educating, developing policies, monitoring progress, researching infusion pumps before purchase or rental, and reporting problems. These strategies involve clinicians, pharmacists, nurses, biomedical engineers, and health information technology professionals and managers. The inclusion of human factors principles and methodologies during design, implementation, and purchasing is one of the most widely accepted approaches [17,24,25,26].

There is limited information available on the role of human factors principles in healthcare purchasing and implementation to improve patient safety. To understand this, we reviewed all the information available in the public procurement system in Spain on infusion pumps, which have been associated with persistent safety problems that can lead to over- or under-infusion and missed or delayed therapy.

## 2. Materials and Methods

To identify the role of human factors and ergonomics in the selection of healthcare technology with high-user-dependence patient safety, represented by infusion pumps, information was retrieved from the Spanish Public Sector Procurement Database (PLACE) (https://contrataciondelestado.es/, accessed on 10th November 2022). This database is used by the Spanish Public Authorities to transparently store their tenders in compliance with the Spanish Law on Public Sector Contracts [27]. This law applies to all contracts for works, work concessions, service concessions, supplies, and services requested by entities belonging to the public sector following the European regulations [28]. PLACE announces more than 11,000 new procurements per month.

For the identification of infusion pump tenders, a search was performed using the Common Procurement Vocabulary [29] (CPV) with the code 33,194,110 (infusion pumps) in the PLACE database between 2002 and 2022. All the records retrieved were reviewed individually by two of the authors, selecting those that included volumetric and syringe infusion pumps and consumables.

All the available documentation for each tender was reviewed, including the supporting memorandum, the tender offer, the technical specifications, the specific administrative clauses, the appointments of the evaluation commissions and the evaluation reports.

From the documentation, information was extracted regarding the dates of publication and tender, the awarding and winning entities, the amounts tendered, the expected duration of the contracts, the number and type of pumps, the number of lots, the technical requirements demanded in each tender and the detailed selection criteria, the professional profiles of the evaluation panels and of the authors of the technical reports, and the planned training in the use of the pumps. In the text of each of the documents, we searched for the words easy, ease, use, usability, intuitive, ergonomics, and human factor.

The records were analyzed and classified in a database to identify the following:Use of human factors-related terms including “ease of use”, “usability”, “human factors”, “ergonomics”, and “intuitive” in the requirements and/or evaluation criteria.Indications of user training requirements or mentions of user training by the bidder.Whether the procurement evaluators had background knowledge and experience in patient safety and/or human factors.Whether human factors methods or principles were mentioned as part of the evaluation.

Numeric data are represented as mean (SD) unless otherwise indicated. Categorical data were compared with the chi-square or Fisher’s exact test. Quantitative data were compared with Student’s t-test as appropriate. A significance level of 0.05 (2-sided) was used for all tests.

## 3. Results

Seventy-three tenders were identified with the infusion pumps’ CPV search codes in the PLACE database, of which forty-four were excluded. The causes for exclusion were emergency tenders or framework agreements (*n* = 20), tenders that did not include syringe or volumetric infusion pumps (*n* = 19), those for only consumables (*n* = 2), those without enough information (*n* = 2), and those for animal equipment (*n* = 1).

The 29 selected tenders were published between July 2015 and October 2022. All the tenders for the procurement of consumable products included the use of infusion pumps during the contract. The overall estimated budget was EUR 30.4 M (the range per procurement was from EUR 46.431 to 12.4 M) for the acquisition of 12.224 pumps and an average duration per procurement of 27 months (range 1 to 60 months). In the 29 selected tenders, 63 batches of pumps with different specifications and selection criteria were identified (19 syringe pump batches and 44 volumetric pump batches).

### 3.1. Requirements and Criteria for Pump Selection

Mandatory technical requirements for the selection of volumetric and syringe infusion pumps identified in the procurement’s batches are listed in Table 1. In a comparison of the requirements of syringes and volumetric pumps, no significant differences were identified, except in the proportion of batches with air system detection.

The overall number of batches with any requirement referring to “ease” was 35 (55.6%), with these mentions related to pump handling (12 cases, 19%), use (9 cases, 14.3%), cleaning (8 cases, 12.7%), programming (6 cases, 9.5%), purging (5 cases, 7.9%), visualizing data (2 cases, 3.2%), placement (1 case, 1.6%), understanding (1 case, 1.6%), and learning (1 case, 1.6%). The number of different requirements referring to “ease” in each batch was 4 in 1 batch (1.6%), 3 in 1 batch (1.6%), 2 in 5 batches (7.9%) and 1 in 28 batches (44.4%). In 18 batches (28.6%), there was at least one requirement related to “intuitive use”. None of the requirements mentioned the terms “usability”, “ergonomics”, or “human factors”.

The scoring criteria for pump selection included in all cases economic and technical aspects with a total of 100 points. The mean points of economic criteria were 46.2 (14.5). Scoring criteria related to the characteristics of pumps are listed in Table 2. Significant differences among syringe and volumetric scoring criteria were detected in pump battery life, pressure monitoring, and relay systems.

Mentions of “ease” in any criteria were present in 23 of the 63 batches (36.5%). They were related to use (12 cases, 19%), programming (12 cases, 19%), purging (8 cases, 12.7%), handling (7 cases, 12.1%), placement (2 cases, 3.2%), and cleaning (1 case, 1.6%). The number of different criteria referring to “ease” in each batch was 1 in 6 batches (9.5%), 2 in 15 batches (23.8%) and 3 in 2 batches (4.8%). Criteria including “intuitiveness” were identified in 10 batches (15.9%) and “usability” in 3 batches (4.3%). No criteria including the terms “ergonomics” or “human factor” were identified.

Training in the use of infusion pumps was a requirement in 22 of the 29 tenders (75.9%). Only in 4 of 63 batches (1.6%) was training considered a criterion for evaluation.

No methodology for the evaluation of any of the requirements or criteria related to ease of use, intuitiveness, or usability was identified. Specifically, there was no mention of human factors evaluation-based principles (i.e., observation, heuristic evaluation, or usability testing) in any of the 63 batches.

### 3.2. Evaluation Procurement Teams

In the analysis of the 29 tenders, only 14 (48.3%) mentioned the professional profile of the members that compose the administrative commissions responsible for the evaluation of the tenders. Overall, in 13 (92.9%), a range of one to seven evaluators with technical non-administrative profiles, such as medical doctors, nurses, or engineers, were included. In eight of them (57.1%), healthcare professionals (nurses or physicians) were included in the commissions. None of the evaluation teams or the technical teams included safety or human factors experts. In seven of the tenders (24.1%), a technical report of the assessment of the bidders by the team of experts with identified profiles was published.

## 4. Discussion

In this review of tenders of public procurement in Spain for supplying 12.224 volumetric and syringe infusion pumps and consumables for a budget of EUR 30.4 M, we identified the main requirements and criteria for selection in 63 different batches and specifically analyzed patient safety- and human factors-related elements. Our data indicate that requirements related to patient safety, such as dose error reduction systems, alarms, or pump blocking systems are frequently considered as requirements or criteria, independently of the type of infusion pump. In this sense, references to the equipment usability-related terms “ease”, “intuitive”, or “usable” were identified in both selection requirements and criteria of the tender documentation in more than half of the cases.

Although concepts related to physical and cognitive ergonomics were present in many of the procurements, no experts in patient safety or human factors were included in the assessment teams of the tenders. Concerning the evaluation of pump usability, no specific methods such as heuristic evaluation or usability testing were identified for any of the procurements.

Medical devices, such as infusion pumps, are used in the healthcare and medical industries to diagnose, prevent, and treat various diseases and disorders. Because medical devices come into direct contact with patients, their construction and design are critical to the effectiveness and safety of treatment. A key element in the successful development of medical devices is the adherence to the principles of ergonomic design and usability. Safety concerns in relation to user-centered design have been analyzed for a variety of devices and systems including infusion pumps [4,5,6].

Since infusion pumps are involved in adverse events, the FDA and other international agencies and associations have launched several initiatives focused on these devices to improve patient safety, including purchasing procurement strategies. The FDA, in its Infusion Pump Initiative, defines several strategies for risk reductions that include formulating and implementing a plan to evaluate infusion pumps before purchasing or renting [17]. In England, the Medicines and Healthcare Products Regulatory Agency [24] emphasized that procurement decision making should be informed by safety performance and reliability assessments. In Canada, the Western Canada Human Factors Collaborative [25] demonstrates that when human factors evaluations are incorporated into procurement activities, procurement committees are better informed, so the chosen devices, equipment, and technologies are more usable, effective, and safer for patients and end-users. This guidance provides comprehensive recommendations on how one may integrate human factor evaluations into procurement processes. Due to limitations such as the availability of human factors experts, time, and multiple procurement efforts running concurrently, choices must be made as to which procurement tasks could include a human factors evaluation. One should prioritize categories or groups of medical devices, equipment, and technology that would be of high significance for the inclusion of human factors evaluation(s) in their procurement process due to known usability and patient safety hazards, including fluid delivery systems as one of the highest priority devices. Other devices prioritized computerized information systems, life-supporting equipment, and surgical devices.

Despite the mentioned recommendations, some data suggest that patient safety is not usually considered a relevant driver of healthcare technology procurement [30]. After detecting several safety incidents related to infusion pumps, the Healthcare Safety Investigation Branch [31] launched an investigation into the NHS to understand the emerging risks and barriers to the safe introduction of the technology and how data may help to demonstrate effectiveness. This investigation found that the procurement of smart pump technology is not primarily driven by the need for smart functionality and was not subjected to a risk assessment or requirement analysis. One of the conclusions of this agency was the need to reinforce that when selecting smart pump devices, it is important to consider how this is likely to impact practice.

Currently, public procurement in Europe is an essential vehicle for implementing government policies and meeting national strategic objectives, as well-functioning public procurement markets contribute to improving the competitiveness of quality service strategies [32]. New strategies are needed to ensure that public procurement addresses existing social challenges such as environmental protection and sustainable consumption and production [33]. Patient safety is a high priority in modern healthcare systems, with the indirect costs of harm running into the trillions in USD each year [1], and its promotion through public procurement can contribute to significant financial savings in reducing patient harm and, more importantly, to better patient outcomes. Recent recommendations to improve the safety of infusion pumps reinforce the need to involve large organizational purchasers of these technologies as they can influence infusion devices and management system design with manufacturers [9].

Several reports have been published about human factors evaluation, specifically of infusion pumps, related to the comparison of different equipment [34], for improving existing designs [11] or for supporting new designs [35,36]. However, human factors evaluation is not a usual practice as a supporting decision tool in medical technology in general [30]. Most medical technology procurement is driven by engineering standards, and the emphasis is on functional requirements rather than those relating to social or organizational needs [37]. Few experiences have been reported in the procurement of volumetric and syringe infusion pumps that incorporate human factors in the decision-making process, pointing out that it adds great value [38,39,40,41,42,43] and demonstrate that human factor and ergonomics evaluation methods, as heuristics or usability evaluation, are affordable as part of the public procurement pathway of infusion pumps with adequate timing, planning, and multidisciplinary teams.

Implementation is a relevant issue of healthcare technology, specifically for improving infusion pump safety. Procurement processes should consider the implementation resources needed, potential barriers and risks to implementation, and the global impact on the organizations during implementation [44]. Our data support previous reports that indicate that the potential of the pumps related to their interoperability and dose error reduction systems is not widely exploited [45,46].

To the best of our knowledge, this study is the first comprehensive analysis of the role of human factors and ergonomics in the public procurement of infusion pumps. Based on the public information available in one country, our data suggest that there is considerable room for improvement in this area, given the lack of a specific methodology for analyzing the safety of medical devices during their selection and implementation. As a specific inclusion of human factors and ergonomics evaluation in the purchase decision of devices with high-risk derived use is advisable, a review of the current evaluation methodology of usability-related requirements and criteria evaluation and the inclusion of new profiles in the teams involved in procurement including patient safety experts should be considered.

## 5. Limitations

This study has some limitations. Although the publication of public procurements is mandatory in Spain and the database used to retrieve procurement information is the official Spanish one, it may not be comprehensive. In addition, there is a lack of uniformity in the schemes and presentation of information in the documents published in the context of public procurements.

It should also be borne in mind that our analysis focuses on data from one country and cannot be automatically extrapolated to another. Although public procurement regulation provides a common basis across Europe [28], there are relevant differences among countries [16]. Future studies should consider other countries or even other public organizations to increase generalizability.

On the other hand, this study only analyzed data from the procurement process and did not correlate accidents that occurred during the actual use of the purchased infusion pumps.

## 6. Conclusions

The deployment of human factors in healthcare organizations implies a global approach and the involvement of different stakeholders, including managers, clinicians, clinical engineers, administrative employees, and human factors experts. Procurement is another opportunity for establishing an adequate implementation strategy, especially when widely used healthcare technology with use-derived hazards such as infusion pumps must be deployed. Public procurement in Europe brings great opportunities for promoting patient safety. Our data indicate that in Spain, the involvement of multidisciplinary teams, considering the human factors perspective in the selection and implementation of technology for improving patient safety, at least in the case of infusion pumps, is an occasion for improving patient safety. This gives healthcare administrators another approach to leading organizational change, considering that patients are as the center of our organizations.

## Figures and Tables

**Table 1 ijerph-20-07179-t001:** Procurement requirements for the selection of syringe and volumetric pumps. Data represent numbers and percentages.

Requirement Aspects	Syringe (*n* = 19)	Volumetric (*n* = 43)	*p*	Overall (63)
Physical Aspects				
Stackability	14 (73.7%)	24 (55.8%)	0.14	38 (61.3%)
Low noise	0	2 (4.7%)	0.48	2 (3.2%)
Weight	15 (78.9%)	32 (72.7%)	0.43	47 (74.6%)
Battery life	14 (73.7%)	33 (75%)	0.57	47 (74.6%)
Alarms and safety systems				
Alarm adjustable volume	5 (26.3%)	18 (41.9%)	0.19	23 (37.1%)
Alarm software	9 (47.4%)	28 (63.6%)	0.19	23 (37.1%)
Pressure alarm	10 (52.6%)	32 (72.7%)	0.1	42 (66.7%)
Obstruction alarm	12 (63.2%)	26 (59.1%)	0.49	38 (60.3%)
Air detection	4 (21.1%)	24 (54.5%)	0.01	28 (44.4.%)
Liquid free-fall prevention	6 (31.6%)	19 (43.2%)	0.28	25 (39.7%)
Safety blocking	6 (31.6%)	14 (31.8%)	0.61	20 (31.7%)
Interface				
Keyboard	5 (26.3%)	12 (28.6%)	0.56	17 (27.9%)
Easy screen	6 (31.6%)	19 (43.2%)	0.28	25 (39.7%)
Screen parameters	12 (63.2%)	25 (56.2%)	0.43	37 (58.7%)
Programming				
Easy management	6 (31.6%)	20 (45.5%)	0.23	26 (41.3%)
Infusion rate	17 (89.5)	41 (93.2%)	0.48	58 (92.1%)
Infusion programming	14 (73.7%)	35 (79.5%)	0.42	49 (77.8%)
Infusion rate medication without interruptions	9 (47.4%)	25 (56.8%)	0.34	34 (54%)
Retrobolus	7 (36.8%)	9 (20.5%)	0.15	16 (25.4%)
Keep vein open	3 (10.5%)	13 (30.2%)	0.09	15 (24.2%)
Drug library	14 (73.7%)	31 (70.5%)	0.52	45 (71.4%)
Pharmacokinetics (target-controlled infusion)	9 (47.4%)	15 (34.1%)	0.24	24 (38.1%)
Interoperability				
Interoperability	10 (52.6%)	14 (31.8%)	0.1	24 (38.1%)
Wi-Fi	2 (10.5%)	5 (11.4%)	0.65	7 (11.1%)

**Table 2 ijerph-20-07179-t002:** Procurement criteria scoring for the selection of syringe and volumetric infusion pumps. Data represent mean of points per criteria (SD).

Criteria	Syringe *n* = 19	Volumetric *n* = 44	*p*	Overall Mean (SD)
Physical Aspects				
Stackability	1.74 (0.67)	1.84 (3.87)	0.51	1.78 (3.54)
Low noise	0.11 (0.47)	0.09 (0.43)	0.81	0.1 (0.43)
Weight	1.50 (3.05)	1.65 (2.89)	0.72	1.56 (2.88)
Battery life	0.17 (0.71)	1.07 (2.88)	0.007	0.78 (2.22)
Alarms and safety systems				
Alarm software	0.67 (1.50)	0.60 (1.58)	0.99	0.6 (1.52)
Noise and lights alarms	0.61 (1.97)	0.40 (1.56)	0.43	0.44 (1.65)
Easy to use	1.72 (2.95)	1.88 (3.59)	0.64	1.78 (3.35)
Drugs library	0.67 (1.41)	1.19 (2.95)	0.14	0.78 (2.44)
Pressure monitoring	7.94 (15.48)	1.70 (5.15)	0.001	3.43 (9.56)
Programming and safety				
Programming software	1.33 (2.89)	1.00 (2.49)	0.49	1.06 (2.56)
Remote monitoring	1.78 (3.14)	1.02 (2.31)	0.06	1.21 (2.55)
Infusion rate	0.67 (2.06)	0.37 (1.23)	0.17	0.46 (1.51)
Infusion volume	1.33 (4.28)	0.95 (3.59)	0.63	1.03 (3.72)
Pharmacokinetics	0.44 (1.46)	0.84 (2.08)	0.12	0.7 (1.89)
Air detection	0.00	0.70 (3.23)	0.06	0.48 (2.68)
Relay system	1.06 (2.48)	0.19 (1.22)	0.001	0.43 (1.69)
Keep vein open	0.28 (1.18)	0.91 (3.41)	0.12	0.7 (2.89)
Interoperability				
Interoperability with other systems	0.56 (2.36)	0.42 (1.93)	0.67	0.44 (2.01)

## Data Availability

Data are available at the Spanish Public Sector Procurement Database (PLACE) (https://contrataciondelestado.es/, accessed on 10 November 2022). with the use of Common Procurement Vocabulary (CPV) with the code 33194110 (infusion pumps) in the PLACE database between 2002 and 2022. The datasets used and analyzed in the current study are available from the corresponding author upon reasonable request.

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
