# Peer review of "Prioritizing Patient Safety: Analysis of the Procurement Process of Infusion Pumps in Spain"

_ijerph, 2023, doi:10.3390/ijerph20247179_

Round 1

Reviewer 1 Report

Comments and Suggestions for Authors

Thank you for giving opportunity to review this paper. Though the authors did a good job, here is some comments to further improve the work:

1) Line 74 the authors said “Several strategies have been proposed for mitigating…etc”. Suggest to add some details on those strategies for better understanding.

2) Please ensure that the problem and purpose statements are explicitly stated.

3) The authors should provide more details to the methods section. It is important to ensure that the methods are robust and suitable for addressing the study purpose.

4) I’m not a statistician, but it seems that the authors did a good job.

5) This research seems original but how can the authors make a significant contribution to the field? Please clarify the impact of your findings. Authors may add a paragraph titled “Study Implications” before the conclusions.

6) Few references are old. Suggest to use more up-to-date references, if possible.

Author Response

Thank you very much for taking the time to review this manuscript. Please find the detailed responses below

1) Line 74 the authors said “Several strategies have been proposed for mitigating…etc”. Suggest to add some details on those strategies for better understanding.

The sentence is corrected to include several strategies that have been proposed for mitigating infusion pump safety problems (proposed by the FDA):

Several strategies have been proposed for mitigating infusion pump safety problems as addressing known problems training and educating, developing policies, monitoring progress, researching infusion pumps before purchase or rental, and reporting problems. The inclusion of human factors principles and methodologies during design, implementation, and purchasing is one of the most widely accepted [17,24]–[26].

2) Please ensure that the problem and purpose statements are explicitly stated.

We add a paragraph at the end of the introduction describing the problem and the statement:

There is limited information available on the role of human factors principles in healthcare purchasing and implementation to improve patient safety. To understand this, we reviewed all the information available in the public procurement system in Spain on infusion pumps, which have been associated with persistent safety problems that can lead to over or under-infusion and missed or delayed therapy.

3) The authors should provide more details to the methods section. It is important to ensure that the methods are robust and suitable for addressing the study purpose.

We add more detailed information to the methods concerning the review of all de documents available:

All the available documentation for each tender was reviewed, including the supporting memorandum, the tender offer, the technical specifications, the specific administrative clauses, the appointments of the evaluation commissions and the evaluation reports.

From the documentation, information was extracted regarding the dates of publication and tender, the awarding and winning entities, the amounts tendered, the expected duration of the contracts, the number and type of pumps, the number of lots, the requirements demanded in each tender and the detailed selection criteria, the professional profiles of the evaluation panels, of those of the authors of the technical reports, and the planned training in the use of the pumps. In the text of each of the documents, we searched for the words easy, ease, use, usability, intuitive, ergonomics, and human factor.

4) I’m not a statistician, but it seems that the authors did a good job.

Thank you for this acknowledgment.

5) This research seems original but how can the authors make a significant contribution to the field? Please clarify the impact of your findings. Authors may add a paragraph titled “Study Implications” before the conclusions.

Accordingly to this recommendation, we add a sentence at the end of the conclusions focused on the implications of the study:

To the best of our knowledge, this study is the first comprehensive analysis of the role of human factors and ergonomics in the public procurement of infusion pumps. Based on the public information available in one country, our data suggest that there is considerable room for improvement in this area, given the lack of a specific methodology for analyzing the safety of medical devices during their selection and implementation. As a specific inclusion of human factors and ergonomics evaluation in the purchase decision of devices with high-risk derived use is advisable, a review of the current evaluation methodology of usability-related requirements and criteria evaluation and the inclusion of new profiles in the teams involved in procurement including patient safety experts should be considered.

6) Few references are old. Suggest to use more up-to-date references, if possible.

Unfortunately we have tried to update the references to include some more recent ones that fit into the scope of the draft and we have not found it.

 Best regards

Reviewer 2 Report

Comments and Suggestions for Authors

This manuscript analyzes the information about the procurement process of infusion pumps in the Spanish government procurement project, finds many possible adverse situations in the procurement process, and puts forward relevant policy improvement suggestions, which is innovative and can arouse the interest of readers.

The title of the manuscript should be modified, it is suggested that associated with specific research content. Infusion pump belongs to the material/consumable purchase, less qualification requirements for procurement, relevant standards and the actual use of weaker demand relationship, so we can get the relevant conclusions of this study. In many countries, the procurement of some special equipment needs to be approved by experts, especially the purchase of some large medical equipment. Some problems mentioned in this study can be easily avoided in this field. So the safety of the procurement process is related to the items purchased.

This study only analyzed the procurement process data, and did not correlate the accidents that occurred in the actual use of these purchased items. Therefore, this study lacks rigor in logic and cannot only analyze the results from the subjective judgment of the purchaser. It is suggested to add relevant research data for re-analysis, or discuss them in detail in the discussion.

Author Response

Thank you very much for taking the time to review this manuscript. Please find the detailed responses below and the corresponding revisions/corrections. 

Following the reviewer's comments, the title is changed to mention infusion pumps. The new title is: Prioritizing Patient Safety: Analysis of Infusion Pump Procurement Process.

 We agree that this study only looks at the purchasing perspective and does not correlate with accidents. This is a limitation of the study and a comment will be added to point this out. The paragraph referring to limitations is rewritten to provide more detailed information

 Limitations

This study has some limitations. Although the publication of public procurements is mandatory in Spain and the database used to retrieve procurement information is the official Spanish one, it may not be comprehensive. In addition, there is a lack of uniformity in the schemes and presentation of information in the documents published in the context of public procurements.

It should also be borne in mind that our analysis focuses on data from one country and cannot be automatically extrapolated to another. Although the regulation of public procurement provides a common basis across Europe, significant differences in the procurement process between countries have been documented. Future studies should consider other countries or even other public organizations to increase generalizability.

On the other hand, this study only analyzed data from the procurement process and did not correlate accidents that occurred during the actual use of the purchased infusion pumps.

Reviewer 3 Report

Comments and Suggestions for Authors

The article is quite interesting and can contribute to the literature and further discussion. Some modifications should be made to improve the quality of the text:

1. I believe that the title of the article (line 2) should be slightly changed because the research presented is about only one country - Spain. A reader who looks at the title of the article may get the mistaken impression that the research presented deals with a problem on a much larger scale than just one country. I suggest adding to the title the phrase: "Based on research in Spain" or something similar.

2. Line 51 - What does the abbreviation "GDP" mean? The abbreviation should be expanded.

3. Line 213 - The abbreviation 'HFE' should be expanded.

4. Limitations should be treated as a separate item. In other words, "Limitations" should be separated from "Discussion" (lines: 223-226).

5. Certainly, a weakness of the article is that the research is limited to one country. Moreover, it is not entirely clear whether the research can be representative (even for Spain). The authors are aware of this. In addition, it is not clear how the studies presented are similar to or different from previous studies and reports (at least some aspects). The authors mention this issue in the Discussion (See: lines 217-222). I think this section could be expanded to indicate specific similarities or differences.

Author Response

Thank you very much for taking the time to review this manuscript. Please find the detailed responses below and the corresponding revisions/corrections
  1. I believe that the title of the article (line 2) should be slightly changed because the research presented is about only one country - Spain. A reader who looks at the title of the article may get the mistaken impression that the research presented deals with a problem on a much larger scale than just one country. I suggest adding to the title the phrase: "Based on research in Spain" or something similar.

RE: According to the comments of the reviewer, the title is modified to mention infusion pumps. The new title is: Prioritizing Patient Safety: Analysis of the Procurement Process on Infusion Pumps in Spain.

  1. Line 51 - What does the abbreviation "GDP" mean? The abbreviation should be expanded.

RE: GDP “means gross domestic product”. The draft is corrected expanding it

  1. Line 213 - The abbreviation 'HFE' should be expanded.

RE: The abbreviation 'HFE' is expanded as “human factor and ergonomics” in the updated draft

  1. Limitations should be treated as a separate item. In other words, "Limitations" should be separated from "Discussion" (lines: 223-226).

RE: Limitations is considered a separate item in the new draft (see below).

  1. Certainly, a weakness of the article is that the research is limited to one country. Moreover, it is not entirely clear whether the research can be representative (even for Spain). The authors are aware of this. In addition, it is not clear how the studies presented are similar to or different from previous studies and reports (at least some aspects). The authors mention this issue in the Discussion (See: lines 217-222). I think this section could be expanded to indicate specific similarities or differences.

RE: Following the reviewer's comments, the limitations section is extended to describe it in more detail, as follows:

Limitations

This study has some limitations. Although the publication of public procurements is mandatory in Spain and the database used to retrieve procurement information is the official Spanish one, it may not be comprehensive. In addition, there is a lack of uniformity in the schemes and presentation of information in the documents published in the context of public procurements.

It should also be borne in mind that our analysis focuses on data from one country and cannot be automatically extrapolated to another. Although the regulation of public procurement provides a common basis across Europe, significant differences in the procurement process between countries have been documented. Future studies should consider other countries or even other public organizations to increase generalizability.

On the other hand, this study only analyzed data from the procurement process and did not correlate accidents that occurred during the actual use of the purchased infusion pumps.

 Best regards